# PARP Inhibitors: Strategic Use and Optimal Management in Ovarian Cancer

**DOI:** 10.3390/cancers16050932

**Published:** 2024-02-25

**Authors:** Nicholas Hirschl, Wildnese Leveque, Julia Granitto, Valia Sammarco, Mervyns Fontillas, Richard T. Penson

**Affiliations:** Massachusetts General Hospital, 55 Fruit St, Boston, MA 02114, USA; nhirschl@mgh.harvard.edu (N.H.); wleveque@mgh.harvard.edu (W.L.); jgranitto@mgb.org (J.G.); vsammarco@mgh.harvard.edu (V.S.); f.mervyns@gmail.com (M.F.)

**Keywords:** olaparib, niraparib, rucaparib, gynecologic, oncology, targeted

## Abstract

**Simple Summary:**

Poly (ADP-ribose) polymerase (PARP) inhibitors have become an essential part of the anticancer armamentarium in ovarian cancer. As maintenence therapy they have been shown to extend survival, and may improve the chance of cure in first line. However, later line treatmewnt is associated with a negative impact on survivaL and limited their use outside trials to first line and secondline in patients with a BRCA mutated cancer.

**Abstract:**

Poly (ADP-ribose) polymerase (PARP) inhibitors have become an established part of the anticancer armamentarium. Discovered in the 1980s, PARP inhibitors (PARPis) were initially developed to exploit the presence of BRCA mutations, which disrupt the homologous recombination repair of deoxyribonucleic acid (DNA) via synthetic lethality, an intrinsic vulnerability caused by the cell’s dependence on other DNA repair mechanisms for which PARP is an essential contributor. PARPi use expanded with the demonstration of clinical benefit when other mechanisms of high-fidelity DNA damage response were present in cancer cells called homologous repair deficiency (HRD). Recently, new data have resulted in the voluntary withdrawal of later-line treatment indications for all the available PARPis used in ovarian cancer because of a negative impact on overall survival (OS). PARPi switch maintenance to consolidate a response to platinum-based therapy is recommended for earlier treatment lines to have the greatest impact on the chance of cure and length of survival. This article reviews the clinical utility of PARPis and how to integrate them into best practices.

## 1. Introduction

The use of PARPis is rapidly evolving, and after an explosion in their use, clinical trials have rapidly informed clinical practice, refining their use to ensure the safest and best care for patients with high-grade serous ovarian and endometrioid cancer. National Comprehensive Cancer Network (NCCN) guidelines (version 2.2023) recommend the use of PARPis as maintenance therapy after the first line (1L) chemotherapy and in patients with *BRCA* (breast cancer-associated gene) mutations in remission with platinum-sensitive disease [1]. There is evidence of benefits in other settings, such as PARPi after PARPi, but sequential use and when to choose PARPi plus the antiangiogenic bevacizumab are more controversial issues. PARPis appear to augment the efficacy of immunotherapy, but this and the study of PARPi combinations with other DNA damaging agents remain an area of active investigation.

PARPs function as an essential part of the “repair crew” for damaged DNA (Figure 1). When deoxyribonucleic acid (DNA) is damaged, cellular proliferation is blocked, and if DNA is not repaired, apoptosis is triggered, destroying the cell and protecting the organism. The first Poly ADP-ribose polymerase (PARP), PARP1, was discovered in 1963 after ATP polymerase was hypothesized as a control on protein synthesis. PARP1 is the most widely studied and abundant of the PARP family. When DNA is damaged, PARP migrates to DNA single-strand breaks, extending a scaffold of PAR as a “Band-Aid” to facilitate DNA repair. An accumulation of DNA SSBs that result in unrepaired stalled replication forks leads to DNA double-strand breaks (DSBs). DNA DSBs normally require the template of a sister chromatid and homologous recombination, a complex function that involves multiple proteins, including *BRCA1* and *BRCA2* [2,3]. The potency of PARP inhibition correlates with the duration of the binding of the drug at the replication fork at DNA repair and is nicely reviewed by Lim and Tan [4]

PARPis prevent PARP-mediated repair, and although initially explored as radiation sanitizers, they also have utility in inducing synthetic lethality. Synthetic lethality occurs when homologous recombination is impaired, such as with inherited or acquired BRCA1 and BRCA2 mutations, and critically disables the cell [5]. Clinically, this is explained to patients as 1. cancer is a “zombie” of mutations, and 2. triggering additional mutations with a PARPi is enough to deliver the lethal blow. A helpful way to think of DNA repair is like a table, with *BRCA* mutations taking out one leg and PARPi taking out another that causes it to topple.

## 2. PARPis in Ovarian Cancer

### 2.1. History of PARPis in Ovarian Cancer

Olaparib (Lynparza™,1800 Concord Pike, WILMINGTON, DE, 19850, USA) was the first PARP inhibitor to be approved in 2014 after the landmark clinical trial, Study 19. This trial provided evidence for the efficacy of olaparib as a maintenance therapy for patients with platinum-sensitive, relapsed serous ovarian cancer, particularly those with *BRCA* mutations, and played a crucial role in the development and regulatory approval of PARPis in the treatment of ovarian cancer [6]. Between August 2008 and February 2010, Study 19 enrolled 265 patients, half of whom received olaparib, while the others received placebo. Among patients with a *BRCA* mutation, the olaparib group demonstrated a significantly longer median progression-free survival (PFS) of 11.2 months versus 4.3 months in the placebo group. When considering all patients, overall survival was similar in both groups: 30 months for those treated with olaparib compared to 28 months with a placebo. However, in patients with BRCA mutations, overall survival was five months better (34.9 months [95% CI 29.2–54.6] vs. 30.2 months [23.1–40.7]; HR 0·62 [95% CI 0.41–0.94] *p* = 0.025), and 11 (15%) of the 74 patients with BRCA mutations received maintenance olaparib for 5 years or more [7]. Apart from fatigue, anemia, and nausea, olaparib was well tolerated.

Rucaparib (Rubraca™,5500 Flatiron Pkwy, Boulder, CO 80301, USA) was the second PARPi to be approved and was granted accelerated approval by the FDA on 19 December 2016 as a monotherapy and received approval for maintenance treatment on 6 April 2018. The most common side effects included nausea, fatigue, transient ALT/AST elevations, and the only grade 3 toxicity was anemia. Adverse reactions led to dose discontinuation in 10% on-study, and MDS/AML was reported in 1%. Clovis’ proprietary BRCA-like DNA HRD signature, developed with Foundation Medicine, appeared to successfully predict which ovarian cancer patients respond to rucaparib therapy [8].

The most recent PARPi to be approved in ovarian cancer is niraparib, GlaxoSmithKline/Tesaro’s PARP inhibitor, Zejula™(1000 Winter St, Waltham, MA 02451, USA). It was granted fast-track designation by the FDA and approved on 27 March 2017 as maintenance treatment for all comers with recurrent epithelial ovarian, fallopian tube, or primary peritoneal cancer who are in complete or partial response to platinum-based chemotherapy and as first-line therapy on 29 April 2020, for all comers.

As genetic testing became more widely available, the use of PARPi in the front line held out the promise of more cures, especially in BRCA mutated tumors [9]. Deficiencies in the homologous recombination DNA repair pathway (HRD), especially in tumors with *BRCA* mutations, became a valuable predictive biomarker [10].

SOLO1 is among the most practice-changing studies to date. It evaluated the use of olaparib as a maintenance therapy in women who had advanced ovarian cancer with a *BRCA* mutation and a complete or partial response to standard platinum-based chemotherapy [11]. Indeed, the study showed that using olaparib specifically in the setting of maintenance therapy significantly extended progression-free survival [12]. The median duration of treatment was two years as planned with olaparib but only 14 months with a placebo. Forty-four percent more participants were alive during follow-up in the olaparib arm [HR 0.55 (95% CI, 0.40 to 0.76)]. This was a remarkable result but did not quite meet the required statistical threshold for significance (*p* < 0.0001) at *p* = 0.0004. At 7 years, 67% of olaparib patients versus 47% of placebo patients were alive [12]. On 19 December 2018, the U.S. Food and Drug Administration (FDA) approved olaparib maintenance after first-line chemotherapy in BRCA-mutated advanced ovarian cancer [13].

In contrast to SOLO1, a broader strategy was pursued with niraparib, which became the first PARPi approved for first-line maintenance for all women with ovarian cancer, regardless of biomarker, and was approved on 29 April 2020 based on the PRIMA study. PRIMA randomized 733 patients, and the median PFS in the HRD population was 22 versus 10 months (HR 0.43; 95% CI: 0.31, 0.59; *p* < 0.0001). The median PFS in the overall population was 13.8 vs. 8.2 (HR 0.62; 95% CI: 0.50, 0.76; *p* < 0.0001). The initial dose is now typically 200 mg once daily; this dose is standard for patients with a baseline body weight of <77 kg and/or a platelet count of <150,000/mm^3^ [14].

Subsequently, on 8 May 2020, the FDA expanded the indication of olaparib to concurrent use with bevacizumab for first-line maintenance treatment of homologous recombination deficient (HRD)-positive advanced ovarian cancer. This is based on PAOLA-1, a randomized, double-blind, international phase III trial in newly diagnosed, advanced, high-grade ovarian cancer comparing olaparib plus bevacizumab to olaparib in a 2:1 ratio as first-line maintenance [15]. In 806 participants, PFS was 22.1 months with olaparib plus bevacizumab and 16.6 months with placebo plus bevacizumab (HR 0.59; 95% CI 0.49–0.72; *p* < 0.001). The hazard ratio (HR) (olaparib group vs. placebo group) for disease progression or death was 0.33 (95% CI, 0.25 to 0.45) in patients with tumors positive for HRD. At ESMO 2022, PAOLA-1 reported a significant improvement in overall survival in HRD-positive tumors (HR 0.62, 95% CI 0.45–0.85; OS at 5 years, 65.5 vs. 48.4%). In the HRP (HR-proficient) group, the median OS was 36.8 months for olaparib–bevacizumab and 40.4 months for bevacizumab (HR  =  1.19 (95% CI, 0.88–1.63)). Sobering the inclination to extend the use of PARPi outside the identification of benefits with HRD [16].

### 2.2. Limitations of PARPis

Like other targeted therapies, resistance to PARPi treatment is common in patients. Several potential mechanisms of resistance have been identified via laboratory in vitro experiments. These mechanisms include the inactivation of specific DNA repair proteins, like 53BP1 and REV7, which lead to the restoration of homologous recombination repair (HRR) and the loss of proteins essential for replication fork stability [17]. Another mechanism involves secondary “revertant” mutations in *BRCA1* or *BRCA2*, which restore the genes’ open reading frames and HRR function, causing resistance to PARPi and platinum-based chemotherapy. This mechanism has been clinically validated as potentially the most important form of PARPi resistance. The selective pressure exerted by PARPi in *BRCA*-defective tumor cells can drive the emergence of resistant clones in advanced cancers. As a result, there is a need for alternative treatment strategies that can either suppress or delay the development of resistant clones. Additionally, since some of these resistance mechanisms affect both PARPi and platinum-based drugs, careful consideration is required when deciding on therapies before and after PARPi treatment for patients [6].

### 2.3. Toxicity

The most common side effects of PARPi are fatigue, anemia, and nausea, with toxicities similar for patients with and without a *BRCA* mutation [18]. In the SOLO2 study, 20% of patients experienced grade 3 or 4 anemia, 4% experienced grade 3 or 4 fatigue, and 3% experienced grade 3 or 4 nausea [19]. Meanwhile, in the NOVA study, 25% of subjects experienced grade 3 or 4 anemia, 8% experienced fatigue, and 11% experienced grade 3 or 4 nausea [20].

While most side effects were mild, the more serious side effects of acute myelogenous leukemia and its prodrome, myelodysplastic syndrome (MDS/AML), are seen approximately twice as often in patients with *BRCA* mutations. In SOLO2, 4% experienced MDS/AML versus 2% in the placebo group [19]. Monitoring of blood pressure is required for niraparib and recommended for all other PARPis.

## 3. Recurrence Therapy for Platinum-Sensitive Diseases

The use of PARPis as maintenance therapy for ovarian cancer following a response to platinum-based therapy has become much more controversial [1]. Since the withdrawal of indications, only olaparib may be used in patients with BRCA mutations, and niraparib is only available for first-line maintenance.

SOLO2 explored the use of olaparib as maintenance therapy in patients with platinum-sensitive relapsed ovarian cancer and a BRCA1/2 mutation in a randomized, placebo-controlled trial. The investigators found a median overall survival benefit of 12.9 months compared to a placebo. Investigator-assessed median progression-free survival was significantly longer with olaparib (19.1 months) than with the placebo (5.5 months) [19]. The median overall survival was 51.7 months (95% CI 41.5–59.1) with olaparib and 38.8 months (31.4–48.6) with placebo (hazard ratio 0·74 [95% CI 0.54–1.00]; *p* = 0.054), unadjusted for the 38% of patients in the placebo group who received subsequent PARP inhibitor therapy [20,21]. 

NOVA Exploratory Analysis showed a statistically significant difference in PFS between niraparib and placebo for maintenance therapy in recurrent ovarian cancer, analyzed with sub-groups comprising HRD-positive BRCA, HRD+ BRCA-wt, and HRD- [20]. The OS rate, however, was found to be statistically insignificant. However, the duration of chemotherapy-free interval and time to the first subsequent treatment favored niraparib and was statistically significant. Patients in the niraparib group had a significantly longer median duration of progression-free survival than those in the placebo group [20].

ARIEL4 displayed median progression-free survival was 7·4 months (95% CI 7.3–9.1) in the rucaparib group versus 5·7 months (5.5–7.3) in the chemotherapy group (hazard ratio [HR] 0.64 [95% CI 0.49–0.84]; *p* = 0.0010). In addition, further analysis of a subset of patients with reversion mutation showed that rucaparib performed significantly worse than the placebo HR of 2.77 [22].

## 4. Overcoming PARPi Resistance with Combinations

Several promising strategies are being pursued.

### 4.1. Combating PARPi Resistance

Targeting alternate DNA repair mechanisms such as the replication fork stabilizing or DNA damage checkpoint proteins ataxia telangiectasia and Rad3-related kinase (ATR) and ataxia-telangiectasia mutated (ATM) are attractive. The combination of olaparib and the ATRi, ceralasertib, demonstrated an objective response rate (ORR) of 50% and activity in patients with recurrent HRD high-grade serous ovarian cancer (HGSOC) who have progressed on prior PARPis [23]. Olaparib and alpelisib, a PI3K inhibitor that is also part of the homologous recombination repair pathway, reported an ORR of 36% [24]. However, overlapping toxicities are likely to limit our ability to combine these sorts of agents with narrow therapeutic windows [25].

Combining PARPis and immune checkpoint blockade (ICB) shows promise in overcoming PARPi resistance by exploiting synthetic lethality with very different approaches [26]. This combination appears to offer extending efficacy beyond the vulnerability of HRD [27]. However, in a trial combining niraparib and pembrolizumab for platinum-resistant ovarian cancer, the ORR was only 18% [26].

### 4.2. Reversion Mutations, PARPi Resistance, and Rechallenge

New innovations in cancer diagnostic techniques have made the detection of specific mutations and indications of resistance more effective while being less burdensome on patients. Crucially, the use of “liquid biopsies”, circulating tumor DNA (ctDNA), and circulating tumor cells (CTCs) provide a non-invasive method for detecting mutations, tumor burden and predicting recurrence. Liquid biopsies provide profound biological insights and useful predictions that increasingly inform treatment choices [28]. For example, BRCA reversion mutations are readily detected using a cell-free DNA sample [29]. Using ctDNA also allows for the monitoring of treatment responses, both post-operatively and during systemic therapies [28]. 

Only one trial has reported on rechallenging with a PARPi in patients with platinum-sensitive relapsed ovarian cancer: the OReO/ENGOT-ov38 trial. Although it is the first randomized placebo-controlled trial to evaluate this approach, it is small (*n* = 146) and has not yet reported on overall survival. OReO demonstrated a statistically significant improvement in PFS, although the authors concede this was modest (PFS 1.5 months better in the BRCA-mutated cohort and 2.5 months better in the non-BRCA-mutated cohort). The greatest benefit was in patients with *BRCA* mutations who were PARPi-naïve. Interestingly, approximately one-third appeared to obtain no benefit, and 10% had a durable response [30]. Reversion mutations are being evaluated to see if this is the main predictor of benefit from PARPi rechallenge in BRCA mutated tumors.

Further research is required to inform the future and optimal use of PARPi with anticipation that they will find an expanded role in augmenting immunotherapy in combinations with other DNA Damage Response interacting agents. This is well reviewed by Giannini et al., but the article’s focus is on the exploration of PARPi in the setting of platinum-resistant disease, which should only be pursued in clinical trials [31].

## 5. Voluntary Withdrawal of Indications

A growing awareness of detrimental effects from PARPi has grown with the review of data from later line use, and in 2022, three voluntary withdrawals of indications announced a significant reduction in the use of PARPi with a remarkable consistent observation in all three PARPis, niraparib, rucaparib, and olaparib. Direct Healthcare Professional Communications (DHPCs) were issued on 10 June 2022 by Clovis Oncology for rucaparib use in the second line and later (2L+) based on ARIEL4 [22,32], 10 August 2022 by AstraZeneca for olaparib based on SOLO3 for 3L+ treatment of gBRCAm tumors [33,34,35], and 20 November 2022 by GlaxoSmithKline for niraparib based on NOVA with an OS disadvantage in 2L+ [36].

Figure 2 compares studies in different PARPi use settings. In the setting of niraparib maintenance (sMx), overall survival (OS) declined after two or more lines of prior therapy for the non-gBRCA mutated HRD-positive subgroup. The median OS was 37 versus 41 months (HR = 1.32 [95% CI 0.84, 2.06]). The ARIEL4 study, shown in the rucaparib treatment setting, for patients with two or more lines of prior therapy had a median OS of 19 months versus 25 months (HR of 1.31 [95% CI 1.00, 1.73]) *p* = 0.0507 [22]. For patients that had three or more prior lines of therapy in the SOLO3 study, treatment of gBRCA mutated median OS was 30 months versus 39 months (HR of 1.33 [95% CI 0.84, 2.18]) [33].

What is clear is that these observations are remarkably consistent; what is less clear is whether the disadvantage comes from BRCA reversion, compromised bone marrow tolerability of subsequent chemotherapy, accelerated mutation and DNA instability, or a direct toxic effect with increased MDS/AML.

## 6. Evolving Clinical Practice

There is a high level of evidence and uniform consensus about the clinical benefit of frontline therapy. However, due to clinician familiarity, there is variation in which agent is most commonly used. Moreover, decisions about sequencing with bevacizumab become more complex (Figure 3). In patients with platinum-sensitive disease, only patients with *BRCA* mutated tumors and who have completed two lines of platinum-based chemotherapy can receive a PARPi, and only olaparib is available in that setting. Caution should be taken when using maintenance PARPi for longer than 24 months, as prior treatment, the patient’s age, and duration of PARPi use appear to be the most powerful predictors of risk of MDS/AML [37]. Selection of therapy for potentially platinum sensitive recurrences now depends on more than simply the time since prior platinum and number of prior lines, as the histology and genetic drives are at least as important [38]. We recommend following standardized protocols, such as the NCCN guidelines, for the selection of PARPis and the duration of their usage (https://www.nccn.org/guidelines/nccn-guidelines) (accessed on 11 February 2024).

## 7. Conclusions

PARPis remain essential options in patients with ovarian cancer with promising activity but diminishing returns later in the clinical course of the disease. These drugs are an essential part of frontline therapy and remain a favorable maintenance option for cancer patients with platinum-sensitive recurrent disease, especially in those patients with a *BRCA* mutation. The introduction of PARPis in *BRCA* mutant patients has transformed clinical practice, but more data are needed on overall survival in ongoing trials to ensure optimal use and more creative clinical trials are required to extend the use of this exciting class of agents.

## 8. Limitations

This article offers a short and succinct explanation of the optimal use of PARP inhibitors in high-grade serous ovarian and endometrioid cancer for clinical minds involved in patient care. By synthesizing data from past research studies on this topic, the authors present a brief but comprehensive overview of best practices for the use of PARPis; no new data are presented here.

## Figures and Tables

**Figure 1 cancers-16-00932-f001:**
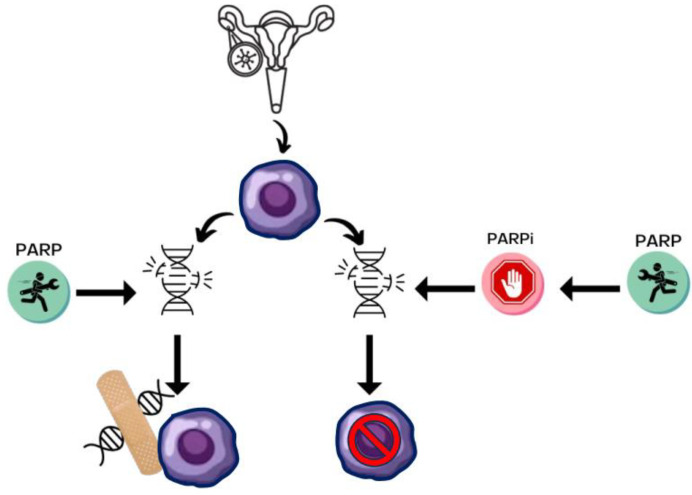
PARPi mechanism of action. Ovarian cancers originate in the fimbria of the fallopian tube and ovary; high grade serous cancers have damaged DNA; PARP is an essential part of repairing DNA; and PARPi prevent repair.

**Figure 2 cancers-16-00932-f002:**
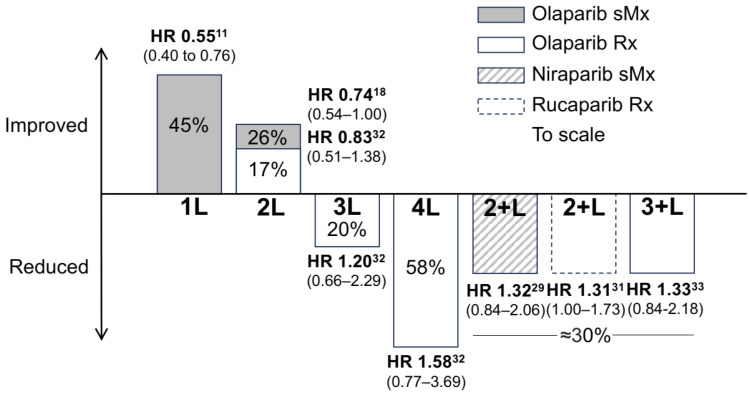
Overall survival impact of PARPi. Hazard ratios (HR) translated into % change in overall suryival. sMx switch maintenance, Rx treatment: 1L first line; 2L second line; 3L third line; 4L fourth line; 2+L two or more prior lines: 3+L three or more prior lines; % percentage.

**Figure 3 cancers-16-00932-f003:**
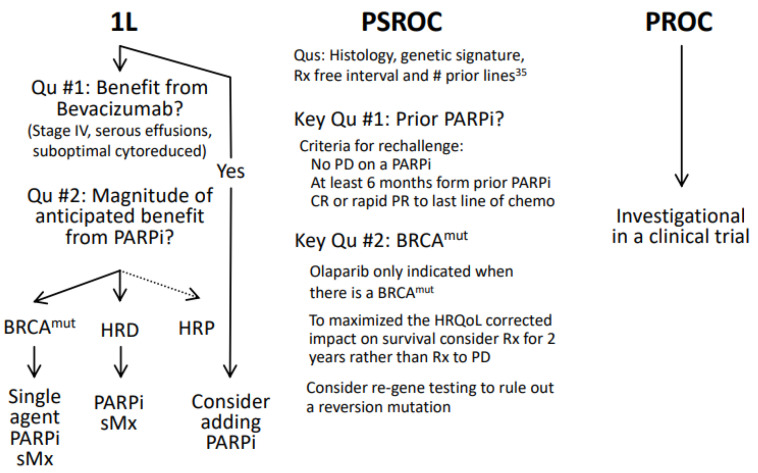
Therapeutic algorithms. Legend: 1L first line; PSROC potentially platinum sensitive recurrent ovarian cancer; PRO potentially platinum resistant ovarian cancer; sMx switch maintenance; ^mut^ mutation.

## Data Availability

Data is contained within the article. The data presented in this study are available via the references listed below.

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
