# Peer review of "PARP Inhibitors: Strategic Use and Optimal Management in Ovarian Cancer"

_cancers, 2024, doi:10.3390/cancers16050932_

Round 1
Reviewer 1 Report
Comments and Suggestions for Authors
The authors summarize the clinical results obtained in recent years with PARP inhibitors (PARPi). The novelty, compared to previous review works on PARPi, concerns the use of these drugs in the subsequent lines of treatment. The work is overall well written, clear, and flowing. However, the last part of the review (starting from paragraph 3 and including figure 3) are less clear and I would ask the authors to review them and explain them better.
Listed below are some minor suggestions for improving the text.
· You should try to put PARP inhibitors with its acronym (PARPis) at the beginning of the text and then always use the acronym.
· By PARP do you mean all parylating PARPs or PARP1, it is not clear in the text. (e.g., Line 40).
· The same acronym is used with two different meanings: homologous recombination DNA repair pathway (HRD) and homologous repair deficiency (HRD). For homologous recombination repair, you use also HRR. Then, Homologous recombination deficient is repeated several times in full in the text instead of using the acronym (e.g., line 224).
· Line 120. Explain HRP in full.
· Line 126. I do not understand how activation of DNA repair proteins leads to restoration of homologous recombination repair (HRR). Maybe you mean revertant mutations like BRCA? Furthermore, at least one bibliographic reference on PARPi resistance should be added.
· Line 142. “…myelodysplastic syndrome (MDS/AML), are seen approximately twice as often in patients with BRCA mutations.” This sentence has no reference. It would be necessary to show data from a clinical study in which there is a comparison of patients treated with cisplatin (well known to induce secondary leukemia) compared to patients treated with cisplatin and olaparib.
· Paragraph 3.1. The authors write that niraparib is approved if deleterious gBRCA mutations are suspected. Are they talking about ovarian cancer? As far as I know niraparib is approved in women newly diagnosed or relapsed with advanced cancer in whom the cancer has shrunk or disappeared with a platinum-based medicines. Please, also check carefully for olaparib and rucaparib, however there are no limitations referring only to the treatment of patients with germline mutations (gBRCA).
· Paragraph 3.1.1. “Evidence Supporting Current Uses of PARPis in Platinum Sensitive Relapsed Ovarian Cancer” remains inconclusive. It should be commented (and especially the last results). Perhaps the entire paragraph could be merged with the previous one.
· Line 164. There is something wrong in the sentence: “...analyzed with sub-groups comprising of HRD Positive BRCA. HRD+ BRCA-wt and HRD-[19].”
· Line 181. HGSOC it should also be written in full as high-grade serous ovarian cancer.
· Write what the acronyms mean in the legend of figure 2.
· In paragraph 5 can you specify whether the patients treated with PARPi in the studies responded to cisplatin?
Author Response
Response to Reviewer 1
The last part of the review has been revised for clarity.
The full-length words for the abbreviation, PARPis is in the abstract.
Homologous recombination deficient has been replaced by HRD where redundant, though we do not find this in line 224.
HRP is now written in full.
In Line 126 we have added an explanation of how activation of DNA repair proteins leads to restoration of homologous recombination repair (HRR). We have added the Patch reference on PARPi resistance. (Ref: Patch, A.M.; Christie, E.L.; Etemadmoghadam, D.; Garsed, D.W.; George, J.; Fereday, S.; Nones, K.; Cowin, P.; Kathryn Alsop, K.; Bailey, P.J.; et al. Whole-genome characterization of chemoresistant ovarian cancer. Nature. 2015;521(7553):489-94.
Re: Line 142 and MDS/AML incidence, we have added the SOLO2 and NOVA studies explaining this with the actual numbers.
We have clarified that niraparib is only approved for 1L sMx.
The recommendation that Olaparib only be used in recurrence for patients with germline mutations (gBRCA) is from AstraZeneca: https://www.lynparzahcp.com/patient-support.html?source=LYN_N_H_2218&umedium=cpc&uadpub=google&ucampaign=gs_lynparzaHCPOneLyn_branded_generic_resources_2024_hcp&ucreative=gs_branded_generic_support_agnostic_ph&uplace=olaparibprogram&outcome=hcp&cmpid=1&gad_source=1&gclid=CjwKCAiAlcyuBhBnEiwAOGZ2S05NtTX5pN_qaXneNFbn9A-nVX1Xq9ONJTdgF-3o7z6nv7jTiwDgJhoCeIkQAvD_BwE&gclsrc=aw.ds
As suggested Paragraph 3.1.1., has been merged with the previous one.
Line 164 has been revised.
HGSOC has been written in full as high-grade serous ovarian cancer in Line 180.
The acronyms in the legend of figure 2 have been written out in full.
It is not possible to be specific about response to cisplatin, as protocols were not consistently specific about the interchangeability of platinum agents, that are generally considered equivalently effective, though with different toxicity profiles.
Reviewer 2 Report
Comments and Suggestions for Authors
Very well written review article that is easy to ready and summarizes the history of PARP inhibitors very well.
Minor points
- Authors should discuss the main differences between PARP inhibitors discussed (Olaparib, Niraparib, Rucaparib) eg their PARP trapping efficiency.
- The authors primarily discuss PARP for the use in Ovarian Cancer. This should be reflected in the title. In addition, the authors should discuss the benefits/weaknesses of PARP inhibitors for the different types of Ovarian cancer ie low grade versus mucinous versus high grade.
- What is the frequency of PARP mutations in ovarian cancer?
Author Response
Response to Reviewer 2
Information about PARP trapping efficacy has been added using a nice review, Lim, J.S.; and Tan, D.S. Cancers. 2017;9(8):pii:E109.
Ovarian Cancer has been added to the title.
A comment about what is considered ‘ovarian cancer’ is really high grade serous or endometrioid, and not low grade or rarer types, especially mucinous has been added.
Although resistance mechanisms is outside the scope of this aricle we have added comment to the effect that PARP mutations are very rare in contrast to mutations in DDR as the major cause of resistance post-PARPi in ovarian cancer (Ref: Patch, A.M.; Christie, E.L.; Etemadmoghadam, D.; Garsed, D.W.; George, J.; Fereday, S.; Nones, K.; Cowin, P.; Kathryn Alsop, K.; Bailey, P.J.; et al. Whole-genome characterization of chemoresistant ovarian cancer. Nature. 2015;521(7553):489-94.
Reviewer 3 Report
Comments and Suggestions for Authors
In this article authors reviewed use of PARPis as a frontline therapy and as maintenance therapy for platinum-sensitive ovarian cancer. Authors discussed about various inhibitors their clinical study results and overcoming resistance. Overall, I agree with authors thoughts on utilizing PARPis in treatment of ovarian cancer. May be authors can comment on developing standardized protocols for selection of PARPis for therapies and duration of their usage. Although toxicity from these drugs is minimal, prolonged usage in older population with BRCA mutations have risk of MDS/AML. I have minor corrections, in most places’ citations are missing, please revise that. Also, on line 120 HRD is misprinted as HRP.
Author Response
Response to Reviewer 3
We have added text to support the use of standardized protocols, such as the NCCN guidelines for selection of PARPis and the duration of their usage.
Line 120 HRP is corrected to HRD.
Reviewer 4 Report
Comments and Suggestions for Authors
I read with great interest the manuscript, which falls within the aim of this Journal and offers a high-quality overview of the topic.
The abstract perfectly summarizes the contents of the manuscript. The introduction is satisfactory.
The methodology is accurate and conclusions are supported by the data analysis. The figures are clear and interesting.
Although the manuscript can be considered already of high quality, I would suggest taking into account the following minor recommendations:
- I suggest another round of language revision, in order to correct a few typos and improve readability.
- Considering the topic analyzed and state of the art in literature, the authors could extend and improve the discussion by evaluating and citing current evidence about possible other target therapeutic strategies for patients with ovarian cancer. I would be glad if the authors discuss this important point, referring to PMID: 37314974
-What is already known on this subject?
-What do the results of this study add?
-What are the implications of these findings for clinical practice and/or further research? It is important to report the results obtained by the authors in the context of clinical practice and to adequately highlight what contribution this study adds to the literature already existing on the topic and to future study perspectives.
- The authors have not adequately highlighted the strengths and limitations of their study. I suggest better specifying these points.
Author Response
Response to Reviewer 4
We have reviewed the manuscript and corrected typos and improved readability.
The review of other targeted therapeutic strategies for patients with ovarian cancer is beyond the scope of the article.
We have added the cited paper: Giannini, A.; Di Dio, C.; Di Donato, V.; D'oria, O.; Salerno, M.G.; Capalbo, G.; Cuccu, I.; Perniola, G.; Muzii, L.;Giorgio Bogani, G. PARP Inhibitors in Newly Diagnosed and Recurrent Ovarian Cancer Am J Clin Oncol. 2023 Sep 1;46(9):414-419., but added the comment that the article’s focus on the exploration of PARP inhibitors in the setting of platinum-resistant disease should only be pursued in clinical trials.
Discussion of future and further research has been expanded.
A section on the review’s strengths and limitations has been added.